# Dimerization of ADAR1 modulates site-specificity of RNA editing

Allegra Mboukou[1,3], Vinod Rajendra[2,3], Serafina Messmer ®[2], Therese C. Mandl[2], Marjorie Catala[1], Carine Tisné[1], Michael F. Jantsch ®[2] ✉ & Pierre Barraud ®[1] ✉

Adenosine-to-inosine editing is catalyzed by adenosine deaminases acting on RNA (ADARs) in double-stranded RNA (dsRNA) regions. Although three ADARs exist in mammals, ADAR1 is responsible for the vast majority of the editing events and acts on thousands of sites in the human transcriptome. ADAR1 has been proposed to form a stable homodimer and dimerization is suggested to be important for editing activity. In the absence of a structural basis for the dimerization of ADAR1, and without a way to prevent dimer formation, the effect of dimerization on enzyme activity or site specificity has remained elusive. Here, we report on the structural analysis of the third double-stranded RNA-binding domain of ADAR1 (dsRBD3), which reveals stable dimer formation through a large inter-domain interface. Exploiting these structural insights, we engineered an interface-mutant disrupting ADAR1-dsRBD3 dimerization. Notably, dimerization disruption did not abrogate ADAR1 editing activity but intricately affected editing efficiency at selected sites. This suggests a complex role for dimerization in the selection of editing sites by ADARs, and makes dimerization a potential target for modulating ADAR1 editing activity.

Adenosine deaminases acting on RNA (ADARs) convert adenosines (A) to inosines (I) in double-stranded endogenous and viral RNAs. Inosines are interpreted as guanosines by most cellular machineries and therefore A-I conversions can recode mRNAs but can also affect the folding or antigenicity of RNAs. Two genes that produce active ADAR proteins can be found in mammals, *Adar* and *Adarb1*[1]. *Adarb1* gives rise to the ADAR2 protein which localizes to nuclei and is mainly expressed in the nervous system, the cardiovascular system and the gastrointestinal tract[2,3]. In mice, loss of ADAR2 leads to lethality around day 21 which can be rescued by pre-editing one of its key substrates, the glutamate receptor subunit *Gria2*[4].

*Adar* encodes the ADAR1 protein which is expressed in all tissues and comes in two versions. A 150 kDa protein (ADAR1 p150) is expressed from an interferon inducible promoter and is mainly localized to the cytoplasm. A constitutively expressed ADAR1 p110 version lacks the amino terminus of p150 and is mainly localized to the nucleus[5,6]. Both ADAR1 p110 and p150 share a C-terminal deaminase domain and three double-stranded RNA-binding domains (dsRBDs) required for substrate binding. A nuclear localization signal (NLS) made of two modules is flanking the third dsRBD in both ADAR1 isoforms[7]. The amino terminus of ADAR1 p150 harbours two Z-DNA binding domains and a nuclear export signal[8,9]. Interestingly, both ADAR1 p150 and p110 can shuttle between the nucleus and the cytoplasm although the steady state localization of p150 is cytoplasmic and that of p110 is nuclear[10,11].

In mice, loss of ADAR1 p150 leads to embryonic lethality while loss of ADAR1 p110 leads to post-natal runted appearance[12]. Interestingly, specific mutations within ADAR1 p150 suggest that different domains contribute to different pathways and phenotypes[13]. Loss of RNA editing activates the innate immune sensor MDA5 leading to interferon signalling[14–16]. The dsRBDs seemingly compete for dsRNA binding with PKR and prevent its activation[17,18]. The ZBDs of ADAR1 p150 have been

[1]Expression génétique microbienne, Université Paris Cité, CNRS, Institut de biologie physico-chimique, Paris, France. [2]Division of Cell and Developmental Biology, Center for Anatomy and Cell Biology, Medical University of Vienna, Vienna, Austria. [3]These authors contributed equally: Allegra Mboukou, Vinod Rajendra. ✉e-mail: michael.jantsch@meduniwien.ac.at; pierre.barraud@cnrs.fr

shown to prevent the activation of ZBP1, another Z-RNA binding protein that leads to necroptosis upon Z-RNA binding[19–21].

Upon binding to double-stranded RNA substrates, the catalytic domain of ADARs flips the substrate adenosine out of its double-stranded context to allow the hydrolytic deamination to occur[22]. As base-flipping is inhibited by base-pairing, ADARs act most efficiently on unpaired adenosines within a certain sequence context[23]. Still, neither dsRBDs nor the catalytic domain show a strong sequence specificity and consequently extended stretches of dsRNA can be edited rather promiscuously. Interestingly, however, RNA-editing events that lead to protein recoding can occur very site-specifically. Structural features of dsRNA substrates seemingly help to correctly position the dsRBDs and the catalytic domain of ADARs. Additionally, dimer formation of ADARs has repeatedly been observed and was discussed as a way to facilitate correct substrate recognition[24–26]. A crystal structure of ADAR2-dsRBD2 and deaminase domains in complex with RNA revealed that this protein can form an asymmetric dimer via its catalytic deaminase domain[27]. Similarly, *Drosophila* ADAR can form dimers albeit via its N-terminal region[28]. Dimer formation of mammalian ADAR1 proteins has also been demonstrated by co-IP and size fractionation experiments[29], as well as fluorescence resonance energy transfer (FRET) experiments in cells transfected with fluorophore-bearing ADAR1[30]. While this dimer formation was shown to be independent of the protein's RNA binding ability, the precise region for ADAR1 homodimerization remained enigmatic[31].

Double-stranded RNA-binding domains have been shown to allow protein-protein interactions and dimer formation[32]. In ADAR1, a region containing its third dsRBD (dsRBD3) had been shown to be essential for dimer formation[33]. In order to identify the exact elements involved in the dimerization of ADAR1, and to characterize the underlying structural aspects, we set out to understand this domain in more detail using X-ray crystallography and solution scattering. Here, we report on the structural analysis of ADAR1-dsRBD3, which reveals dimer formation with a large interdomain interface at the level of the dsRBD's β-sheets. In addition, co-immunoprecipitation and mutational analysis confirm a dsRBD3-mediated dimer formation of ADAR1 in vivo. We show further, that dimer formation affects editing efficiency at selected sites in ADAR1 model substrates, suggesting a complex role for dimerization in the selection of editing sites by ADARs.

## Results

### The folded region of ADAR1-dsRBD3 is sufficient to mediate dimerization

Nishikura and colleagues identified that regions around dsRBD3 of ADAR1 are required for the homodimerization of ADAR1[33]. More precisely, a construct lacking the segment covering residues 725–833 resulted in the loss of interaction with full-length ADAR1. We determined earlier that the folded part of dsRBD3 of ADAR1 consists of residues 716–797[7]. We therefore wanted first to better delineate the elements needed for ADAR1 dimerization and to determine whether the folded domain or the flanking regions, or both, are involved in the interaction. To do so, we produced and purified three constructs of different size, which all encompass at least the folded part of ADAR1-dsRBD3 (i.e. residues 716–797), but with different flanking regions. In addition to a construct covering only the folded portion of dsRBD3,

namely dsRBD3-short (residues 716–797), we generated a dsRBD3-mid construct (residues 708–801), a construct that we previously found to be the minimal ADAR1 construct that retains nuclear localization properties[7], and a dsRBD3-long construct (residues 688–817), which was chosen to contain an additional -15–20 residues on both sides of the dsRBD-mid construct (Supplementary Fig S1A). Purification tags were cleaved off to avoid potential unspecific interactions. All constructs were analysed by small-angle X-ray scattering experiments coupled to size-exclusion chromatography (SEC-SAXS). The scattering data revealed that all three constructs behave as dimeric domains in solution (Table 1, Supplementary Table S1 and Supplementary Fig S1B). These experiments demonstrate that ADAR1 dimerization is an intrinsic property of its third dsRBD that forms a stable dimer in solution on its own, without the need for any additional domain.

### The crystal structure of ADAR1-dsRBD3 reveals a dimerization interface

We determined the crystal structure of the folded entity of ADAR1-dsRBD3 (residues 716–797) and refined it to a resolution of 1.65 Å (Supplementary Fig S2A and Supplementary Table S2). ADAR1-dsRBD3 crystallized in the hexagonal space group P3$_1$21 with two dsRBDs in the asymmetric unit (Fig. 1A). Overall, the structure of the individual dsRBDs corroborate the structure previously solved by nuclear magnetic resonance (NMR) spectroscopy, confirming the classical topology of dsRBDs with the core α1-β1-β2-β3-α2 elements, as well as the additional N-terminal helix α$_N$, important for assembling the bimodular NLS of ADAR1[7]. However, as a consequence of the way NMR structures are calculated, namely by collecting and interpreting a large number of short-range distances, the previously solved solution structure failed at identifying the dimeric state of the domain. This most probably also arose since ADAR1-dsRBD3 forms a symmetric dimer, in which the NMR resonances coming from identical residues in each monomer are perfectly superimposable. On the contrary, in a crystal structure, the overall assembly is readily accessible, which here revealed that the two dsRBDs of the asymmetric unit interact via an extended interface at the level their β-sheet surface (Fig. 1). This interface has a surface area of -575 Å², which corresponds to a total buried area of -1150 Å². The two monomer structures are not perfectly identical as a result of small local variations caused by differences in the packing assembly, and display an r.m.s.d. of 1.19 Å over the entire Cα of the domains (Supplementary Fig. S3). Nevertheless, the monomers are assembled via a pseudo C2-symmetry and form a virtually symmetric dimer (Fig. 1B), that would most probably be truly symmetric without the structural distortions caused by crystal packing. Interactions at the dimer interface involve residues from all β-strands, namely V747 and D748 from β1, K757, V759 and Q761 from β2, and W768, P770, A771 and C773 from β3. Interactions are formed by a combination of electrostatic and hydrophobic contacts. On one edge of the β-sheet, W768 stacks onto the P770 cycle and the side chain Q761 interacts via two hydrogen bonds with the backbone of β3, namely the carbonyl of F769 and the amide of A771 (Fig. 1C). On the other edge, a small hydrophobic core is assembled via the side chains of V747, V759 and C773, and an electrostatic contact is formed between D748 and K757 (Fig. 1D). All contacts are present twice

**Table 1 | Summary of SEC-SAXS data used to derive the molecular weights of ADAR1-dsRBD3 constructs**

| ADAR1-dsRBD3 construct | MW (kDa) | Averaged frames | Rg (Å) | Dmax (Å) | Apparent MW [95%]ᵃ (kDa) |
|---|---|---|---|---|---|
| dsRBD3-long | 14.7 | 235–249 | 25.5 ± 1.0 | 80 | 30.9 [29.2–33.4] |
| dsRBD3-mid | 10.8 | 312–318 | 20.1 ± 0.9 | 67 | 18.7 [17.8–19.6] |
| dsRBD3-short | 9.2 | 299–305 | 19.2 ± 1.1 | 55 | 21.2 [19.6–22.7] |

ᵃThe apparent MW in solution is given as an averaged value as well as an interval of confidence at 95% in brackets, all determined with PRIMUS (ATSAS suite) using default settings.

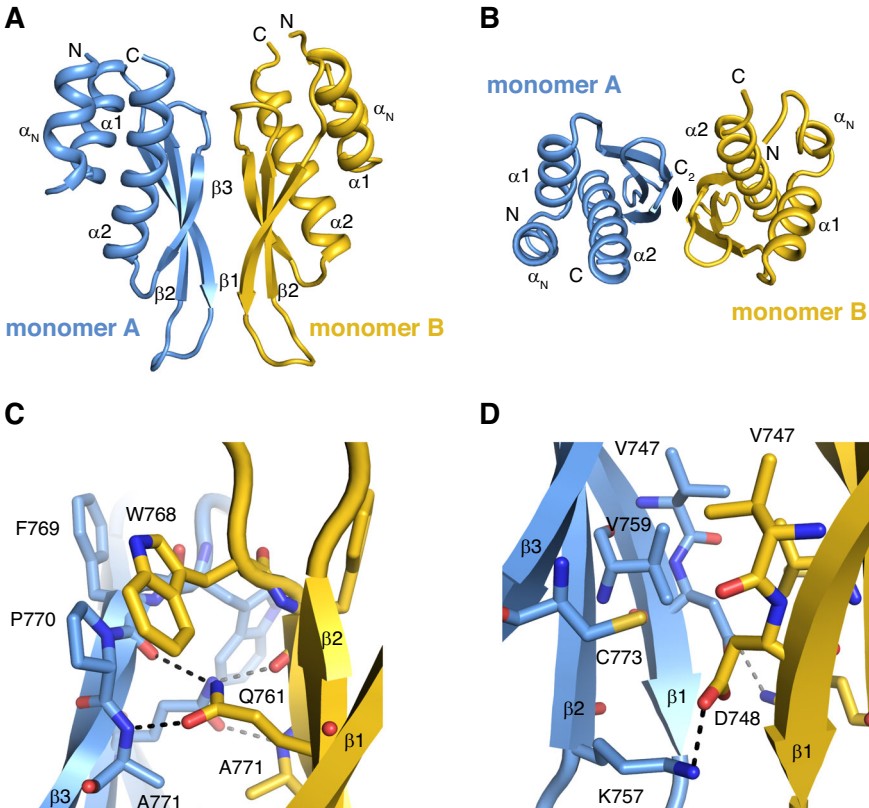

**Fig. 1 | ADAR1-dsRBD3 forms a stable symmetric dimer. A** Overall organization of ADAR1-dsRBD3 dimer. Monomers A and B are displayed in cartoon mode and coloured *in blue* and *in yellow*, respectively. Secondary structure elements are labelled on each monomer. **B** ADAR1-dsRBD3 forms a symmetric dimer. Top view along the $C_2$ symmetry axe. **C** Contacts at the dimer interface between monomer A (*in blue*) and monomer B (*in yellow*) on one edge of the β-sheet. Important residues are shown as sticks. Polar contacts are shown as dashed lines. **D** Contacts at the dimer interface between monomer A (*in blue*) and monomer B (*in yellow*) on the other edge of the β-sheet. Important residues are shown as sticks. Polar contacts are shown as dashed lines. Data have been deposited to the PDB (accession code 7ZJ1).

between each of the monomers owing to the symmetrical nature of the interface.

## The dimer observed in the crystal corresponds to the one existing in solution

Next, to establish that the dimer observed in the crystal structure is also the one that exist in solution, we calculated theoretical scattering curves from the crystal structure of ADAR1-dsRBD3 using CRYSOL and compared them with the experimental SEC-SAXS data obtained with the dsRBD3-short construct. The theoretical curve obtained with the ADAR1-dsRBD3 monomer structure differs substantially from the experimental data (goodness of fit $\chi^2 = 41.8$), but the one obtained with the ADAR1-dsRBD3 dimer is highly similar to the experimental data (goodness of fit $\chi^2 = 1.30$) (Fig. 2). This indicates that the dimeric structure observed in the crystal is also a good representation of ADAR1-dsRBD3 in solution. In addition, we performed ab initio shape reconstructions from the dsRBD-short SEC-SAXS data using independently the DAMMIN and GASBOR programs. These ab initio dummy residue models further confirmed that the solution scattering data obtained in solution with ADAR1-dsRBD3 are perfectly compatible with the dimeric form observed in the crystal structure (Supplementary Fig. S4).

## Dimerization of ADAR1-dsRBD3 is compatible with its binding to dsRNA

Having shown that ADAR1-dsRBD3 dimerizes via its entire β-sheet surface, we were wondering whether this dimerization affects the binding of the domain to dsRNA. We were indeed wondering whether

the strong binding of ADAR1-dsRBD3 to dsRNA that we previously characterized by isothermal titration calorimetry (ITC)[7] occurs with ADAR1-dsRBD3 in its dimeric form. For that, we sought at obtaining structural information to reveal how ADAR1-dsRBD3 interacts with dsRNA. For this purpose, we have set up crystallization assays using small oligonucleotides of 11–13 residues that can base-pair with themselves to form short duplexes and associate with other duplexes via their two-nucleotide overhangs to form long dsRNA helices, a strategy designed to facilitate crystallization[34]. We could obtain well-diffracting crystals only with the longest RNA, e.g. 13-nucleotide long RNA, and refined the structure of the ADAR1-dsRBD3/dsRNA complex to a resolution of 2.8 Å (Supplementary Fig. S2B and Supplementary Table S2). ADAR1-dsRBD3 crystallized in the hexagonal space group P6₁22 with two dsRBDs and three oligonucleotides in the asymmetric unit (Fig. 3A). Within the crystal, the oligonucleotides are assembled into long pseudo-A-form RNA helices through their CG 3′-overhangs (Fig. 3B). Each ADAR1-dsRBD3 monomer binds to such a dsRNA helix at the exact same position, meaning that residues from each dsRBD are forming the same type of contacts with identical nucleotides on two different dsRNA helices. These contacts include mainly non-sequence-specific contacts with the sugar-phosphate backbone of the RNA helices (Fig. 3C). Typically, molecular recognition of dsRNA by dsRBDs occurs via three regions of interaction: helix α1 and the loop between β1 and β2 contact two dsRNA minor grooves at one turn of interval, whereas the N-terminal tip of helix α2 contacts the dsRNA phosphate backbone across the major groove[35]. The binding mode that we observe here is similar to what was observed for other dsRBD-dsRNA complexes[36–44]. In particular, in region 1, helix α1 makes non-sequence-

specific contacts with 2′-OH groups on each RNA strands, namely with U9 and A4, via residues E733 and R736, respectively (Fig. 3C, D). A point worth noting in this region concerns the additional N-terminal helix $\alpha_N$ that also participates in the recognition by making additional contacts with the sugar-phosphate backbone, namely with U8 and G5, via residues R721 and N726, respectively (Fig. 3C, D). In region 2, residue H754 forms non-sequence-specific contacts with 2′-OH groups on each RNA strands, namely with C10 and A3, via its main-chain carbonyl and its imidazole ring Nδ1, respectively. In addition, residue P753 forms a sequence-specific contact with the exocyclic amino group of G2, via its main-chain carbonyl (Fig. 3C, E), as previously reported in many dsRBD-dsRNA structures[35]. Finally, in region 3, lysines K777 and K778 form electrostatic contacts with the phosphate backbone of each RNA strand across the major groove. Namely, K777 interacts with non-bridging oxygen atoms of C12 and G13, via its main-chain amide and side-chain terminal amino group, respectively; and K778 interacts with non-bridging oxygen atoms of C7 and U8 on the other strand, via its side-chain terminal amino group. Additionally, residue Q782 contacts the 3′ bridging oxygen atom of C6 via its side-chain amide group (Fig. 3C, F). Finally, the third lysine of the conserved KKxxK motif, namely K781, is found further away from the RNA sugar-phosphate backbone (>6 Å), but could form a water-mediated contact with a non-bridging oxygen atom of C6, as previously reported[45]. Altogether, apart from the remarkable contribution of helix $\alpha_N$, RNA-binding by individual monomers of ADAR1-dsRBD3 corresponds to the well-defined canonical mode of dsRNA binding.

In addition to the RNA recognition mode by individual monomers, in the case of ADAR1-dsRBD3, the ability to bind to two dsRNA helices as a dimer is of particular interest (Fig. 3A). Importantly, the two dimers observed in the crystal structures of ADAR1-dsRBD3 alone (Fig. 1A), or in complex with dsRNA (Fig. 3A), are completely superimposable (r.m.s.d. of 0.63 Å over the entire Cα of the domains – Supplementary Fig S5). Since the dimerization of ADAR1-dsRBD3 occurs via the β-sheet surface, which is located opposite to the dsRNA-binding surface, the dimerization does not prevent the domain to bind to dsRNA, and each dsRBD monomer can bind to a dsRNA helix (Fig. 3A). This feature gives ADAR1 a remarkable position in terms of substrate RNA-binding (see discussion).

## Mutations at the interface render ADAR1-dsRBD3 monomeric

In order to investigate the importance of the ADAR1 dimeric status for its cellular functions, we sought to design point mutations at the interface of ADAR1-dsRBD3s that would disrupt the domain interactions and result in a monomeric ADAR1. For this, we have based our strategy on the structural comparison of ADAR1-dsRBD3 with an archetypal monomeric dsRBD, i.e. the Xlrbpa-dsRBD2 (pdb code 1di2)[36], a dsRBD we already used in our previous study to design a chimeric ADAR1-dsRBD3/Xlrbpa-dsRBD2 that retained the nuclear localization properties of ADAR1[7]. We thus first superimposed our ADAR1-dsRBD3 structure with that of Xlrbpa-dsRBD2 and derived the corresponding structure-based sequence alignment (Supplementary Fig. S6). Based on these structure and sequence alignments, and considering the main interactions observed at the ADAR1-dsRBD3 interface (Fig. 1C, D), we decided to create a mutated construct with four point-mutations, i.e. V747A and D748Q in strand β1 and W768V and C773S in strand β3, with the aim of disrupting the dimer interface. We expressed and purified this construct and measured its apparent molecular weight with multi-angle laser light scattering coupled to size-exclusion chromatography (SEC-MALLS) and SEC-SAXS. As a control, we also studied the chimeric ADAR1-dsRBD3/Xlrbpa-dsRBD2 construct we previously studied[7]. This construct consists of the entire Xlrbpa-dsRBD2 sequence, flanked by the N- and C-terminal modules forming the bimodular NLS of ADAR1, together with the additional N-terminal helix $\alpha_N$ of ADAR1 and few point mutations to allow the correct accommodation of this helix (Supplementary Fig. S6)[7]. This

construct was chosen since it is expected to be monomeric, similar to Xlrbpa-dsRBD2, while retaining the intracellular localization properties of ADAR1. All constructs used in this part were based on dsRBD3-mid (residues 708–801) that retained the N-terminal purification tag, and were checked to be properly folded using NMR spectroscopy (Supplementary Fig. S7). Scattering data in solution, i.e. SEC-MALLS and SEC-SAXS, confirm that the chimeric ADAR1-dsRBD3/Xlrbpa-dsRBD2 construct is indeed monomeric (Table 2 and Supplementary Fig. S8). Most importantly, the mutated construct at the interface (V747A, D748Q, W768V and C773S) also behaves in solution as a monomer. The estimated MW derived from both SEC-MALLS and SEC-SAXS are matching the expected MW for a monomer and the retention times on the size exclusion column are corresponding to the retention time of the monomeric ADAR1-dsRBD3/Xlrbpa-dsRBD2 chimera and differ from the one of the wild-type ADAR1-dsRBD3, which, as already observed, elutes and behaves as a dimer in solution (Table 2 and Supplementary Fig. S8). We would like to point out that the studied samples display some non-ideal behaviour both in SEC-MALLS and SEC-SAXS, with a certain level of aggregation and probably some interaction with the SEC column, leading to non-symmetrical profiles. While such properties might prevent the use of SEC-SAXS data for accurate molecular modelling, the simple molecular weight estimates derived from these data can confidently be used to estimate whether the constructs are behaving as monomers or dimers in solution.

Having designed a quadruple mutant (V747A, D748Q, W768V and C773S) that renders ADAR1-dsRBD3 monomeric, we have now a molecular tool at hand to investigate the molecular and cellular functions of ADAR1 dimerization. In the following parts, we have used this interface mutant for exploring diverse aspects of ADAR1 biology, namely RNA-binding, cellular localization and editing activity.

## Monomeric ADAR1-dsRBD3 remains competent for RNA-binding and nuclear localization

First, we tested whether disrupting the dimeric interface in ADAR1-dsRBD3 would drastically alter the RNA-binding capacity of the domain. For that, we measured with ITC the dsRNA binding capacity of wild-type dimeric ADAR1-dsRBD3, monomeric ADAR1-dsRBD3 mutant, and monomeric ADAR1-dsRBD3/Xlrbpa-dsRBD2 chimera. We used an RNA duplex of 24-base pairs as a model of a dsRNA helix capable of accommodating various dsRBDs as previously described[7,46]. We showed with ITC titrations (Supplementary Fig. S9), that monomeric ADAR1-dsRBD3 binds dsRNA with a strong affinity ($K_D = 410 \pm 65$ nM), although its RNA-binding occurs with a slightly lower affinity compared with wild-type ADAR1-dsRBD3 ($K_D = 125 \pm 25$ nM). This corresponds to a typical RNA-binding affinity when compared to other dsRBDs[40,42,47,48], and is also similar to the binding affinity measured for the chimeric ADAR1-dsRBD3/Xlrbpa-dsRBD2 construct ($K_D = 500 \pm 55$ nM – Supplementary Fig. S9). Thus, disrupting dimer formation does not prevent ADAR1-dsRBD3 from binding to dsRNA.

As the bimodular NLS of ADAR1 is also flanking dsRBD3[7], we tested whether the dimerization mutation would interfere with nuclear localization. We therefore transfected either full-length FLAG-tagged ADAR1 p110 and ADAR1 p150 wild-type or a version harbouring the dimerization mutations in dsRBD3 into human embryonic kidney 293T (HEK293T) cells. Immunofluorescence staining confirmed that ADAR1 p110 remained nuclear in the presence of the dsRBD mutations preventing dimerization, while ADAR1p150 showed the expected cytoplasmic localization (Supplementary Fig. S10A). Disrupting dimer formation at the level of its atypical NLS does not hinder ADAR1 from being properly delivered by the nucleocytoplasmic transport machinery.

## ADAR1-dsRBD3 mediates dimerization in vivo

In ADAR1, as mentioned above, dimerization had been noticed via both a region flanking dsRBD3 but also the deaminase domain itself[27,29,33].

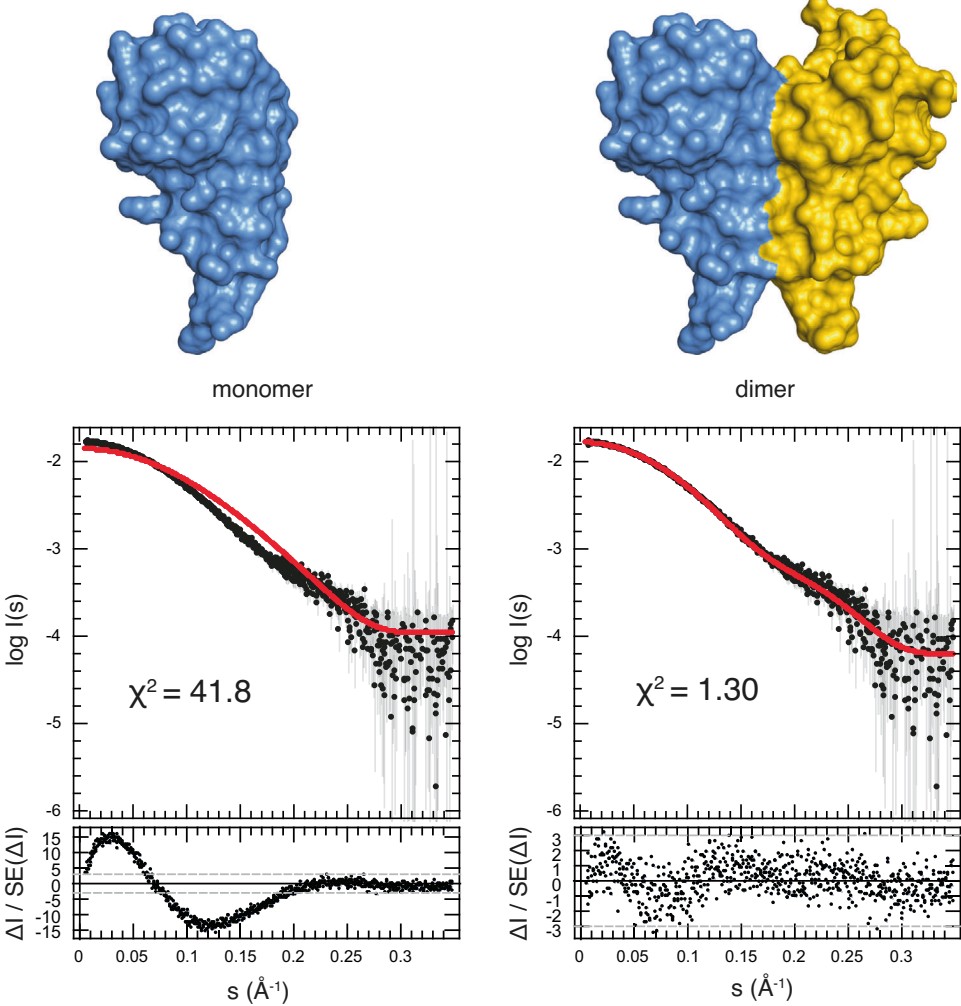

**Fig. 2 | The ADAR1-dsRBD3 dimer observed in the crystal also exists in solution.** SAXS characterization of ADAR1-dsRBD3 (dsRBD3-short construct, residues 716–797). Upper panel: Surface representation of ADAR1-dsRBD3 monomer (*left*) or dimer (*right*) structure determined by crystallography (see Fig. 1). Lower panel: Experimental data are shown as black dots with estimated experimental errors as grey bars. The theoretical scattering fits, as reported by CRYSOL from the crystal structure atomic coordinates of dimeric and monomeric ADAR1-dsRBD3, are shown as red dots. The error-normalized fit residuals are reported below as small black dots. Goodness of fit ($\chi^2$) are reported in each case. Data have been deposited to the SASBDB (accession code SASDVH7).

We therefore tested the ability of dsRBD3 interface mutant for dimer formation both in the presence and absence of the deaminase domain in HEK293T cells using co-immunoprecipitation (co-IP). To do so, full-length wild-type and mutant versions FLAG- and HA- tagged ADAR1 variants were co-expressed in HEK293T cells. Subsequently the FLAG-tagged proteins were precipitated using anti-FLAG coupled magnetic beads and purified complexes were tested for the presence of HA-tagged interaction partners by western blotting. This was performed in the presence of RNases to disrupt any RNA-mediated co-precipitation. All experiments were performed in triplicate and quantified (Fig. 4B, D). As expected, full-length versions showed an interaction irrespective of wild-type or dimerization-mutant dsRBD3, due to the presence of the deaminase domain that also presents self-interaction properties (Fig. 4A, B and Supplementary Fig. S11)[27].

However, when the deaminase domain was deleted, only constructs with an intact dimerization domain in dsRBD3 could copurify, while mutations preventing dimerization of dsRBD3 disrupted dimer formation (Fig. 4C, D and Supplementary Fig. S11). To ensure that deletion of the deaminase domain does not interfere with cellular localization of ADAR1 we also performed immunofluorescence staining of the HA-tagged, delta-deaminase constructs, carrying or lacking

the dimerization mutation. As expected, deletion of deaminase domains both in the wild-type and dsRBD3 interface mutations did not affect their cellular localization (Supplementary Fig. S10A). Together, these results indicate that dsRBD3 of ADAR1 is sufficient to mediate ADAR1 dimerization as shown in the crystal structure.

**Mutations that abolish dsRBD3 dimerization affect editing activity in vivo**

As we had shown that both monomeric and dimeric forms of ADAR1-dsRBD3 remained competent for RNA binding in vitro (Fig. 3 and Supplementary Fig. S9), we wondered whether mutations preventing dsRBD3 dimerization would affect editing activity of either ADAR1 p150 or ADAR1 p110 in vivo. For this purpose, we ectopically expressed FLAG-tagged ADAR1 p150, ADAR1 p110 and the corresponding dsRBD3 interface mutants (i.e. V747A, D748Q, W768V and C773S) in HEK293T cells and measured editing levels at endogenously expressed, previously studied sites[27,49]. Expression of ADAR1 variants was tested by western blotting (Supplementary Fig. S10B). To allow for a direct comparison of multiple editing sites and to detect site-specific editing preferences of wild-type and mutant ADAR versions, several sites were simultaneously amplified from three replicate

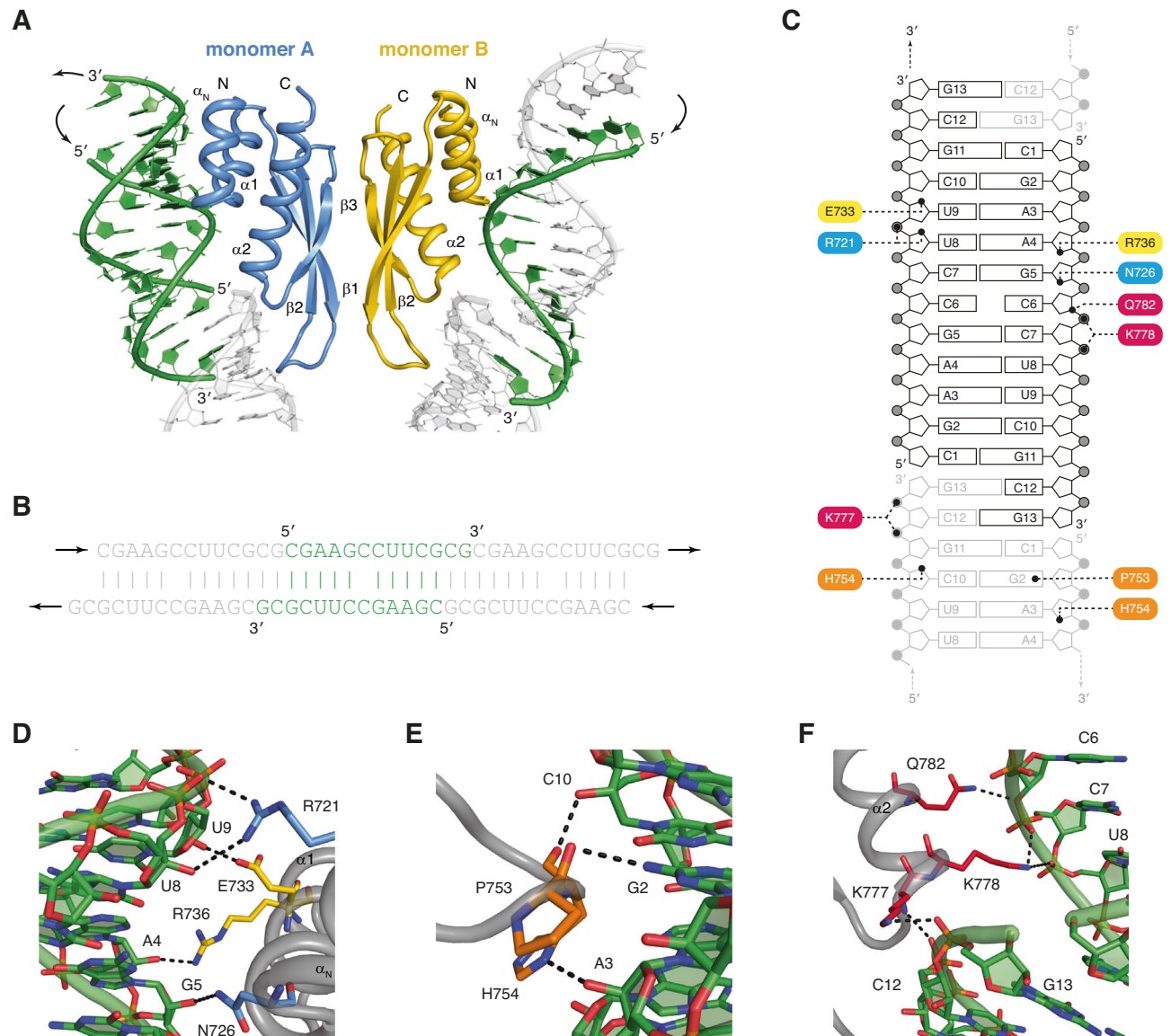

**Fig. 3 | Dimerization of ADAR1-dsRBD3 is compatible with its binding to dsRNA.** **A** Overall organization of the asymmetric unit of ADAR1-dsRBD3:dsRNA crystal structure. Monomers A and B are displayed in cartoon mode and coloured *in blue* and *in yellow*, respectively. Secondary structure elements are labelled on each monomer. dsRNA helices are displayed in green and in grey, for RNA strands within the asymmetric unit or symmetry-related RNA strands, respectively. **B** Schematic representation of the RNA self-assembly. The RNA sequence (5′-CGAAGC-CUUCGCG-3′) contains two-nucleotides 3′-overhangs used to generate a dsRNA by self-hybridization. The central dsRNA block is shown *in green*, and the flanking blocks *in grey*. Arrows indicate the direction of self-assembly that leads to a pseudo-A-form dsRNA helix. **C** Schematic representation of ADAR1-dsRBD3:dsRNA contacts. Dotted lines indicate contacts between ADAR1-dsRBD3 residues and the RNA. Residues from various dsRBD regions are shown with different colours, with residues from helix αN *in blue*, from helix α1 *in yellow*, from the β1-β2 loop *in orange*, and from the N-terminal tip of helix α2 *in red*. **D–F** Detailed views of the three regions of interactions. Polar contacts are displayed as dotted lines. The same colour code is used as in panel C. Data have been deposited to the PDB (accession code 7ZLQ).

complementary DNA (cDNA) preparations and sequenced by Sanger sequencing.

*Azin1* carries two editing sites that are mostly edited by ADAR1 p150[50,51]. Here, site 1 was edited to ~25% by wild-type ADAR1 p150 while editing was almost completely abolished by the dimerization mutant (Fig. 5A). Site 2 showed ~10% editing by the wild-type enzyme and the dimerization mutant showed only 4% editing (Fig. 5B). While this might suggest that the mutations preventing dimerization of dsRBD3 can generally reduce editing, this was not the case for a prominent site in *Gli1*. In the context of ADAR1 p150 the dimerization mutant showed reduced editing from ~60% to ~40% while the same mutation in the context of the ADAR1 p110 isoform did not lead to a significant

reduction in editing (Fig. 5C). To get an even better view on the impact of the dimerization mutant we analysed editing in *Cflar* and *Nicn1*, two targets with inverted short interspersed nuclear elements (SINEs) and therefore harbouring multiple closely-spaced editing sites within the 3′ untranslated region (3′ UTR). Again, editing patterns between both ADAR1 p150, ADAR1 p110 wild-type and the corresponding dsRBD3 interface mutants were investigated. In *Cflar*, editing levels at sites 1 and 2 seemed not affected by loss of dimerization, while editing at site 3 was strongly reduced for the mutant versions both in the context of ADAR1 p150 and ADAR1 p110 (Fig. 5D, E). Sites 4, 5, and 6 are mainly targeted by ADAR1 p110. Here, a total loss of editing was seen at sites 4 and 5 for the dsRBD3 dimerization mutant in the context of ADAR1

**Table 2 | Summary of SEC-MALLS and SEC-SAXS data used to derive the molecular weights of ADAR1-dsRBD3 wild-type and mutant constructs**

| Construct | MW (kDa) | SEC-MALLS | | SEC-SAXS | | | |
|---|---|---|---|---|---|---|---|
| | | Retention time (min) | Apparent MW (kDa) | Retention time (min) | Averaged frames | Rg (Å) | Apparent MW [95%]ᵃ (kDa) |
| ADAR1-dsRBD3 | 12.6 | 8.74 | 24.0 ± 4.3 | 11.39 | 305–325 | 24.4 ± 3.5 | 24.3 [22.7–25.3] |
| ADAR1-dsRBD3 interface mutant | 12.5 | 9.07 | 14.2 ± 1.6 | 11.88 | 334–358 | 21.3 ± 0.6 | 12.0 [10.8–13.1] |
| Chimeric ADAR1-ds3/ Xlrbpa-ds2 | 12.8 | 9.03 | 12.3 ± 0.5 | 11.90 | 333–351 | 20.2 ± 1.0 | 11.2 [10.8–12.4] |

ᵃThe apparent MW in solution is given as an averaged value as well as an interval of confidence at 95% in brackets, all determined with PRIMUS (ATSAS suite) using default settings.

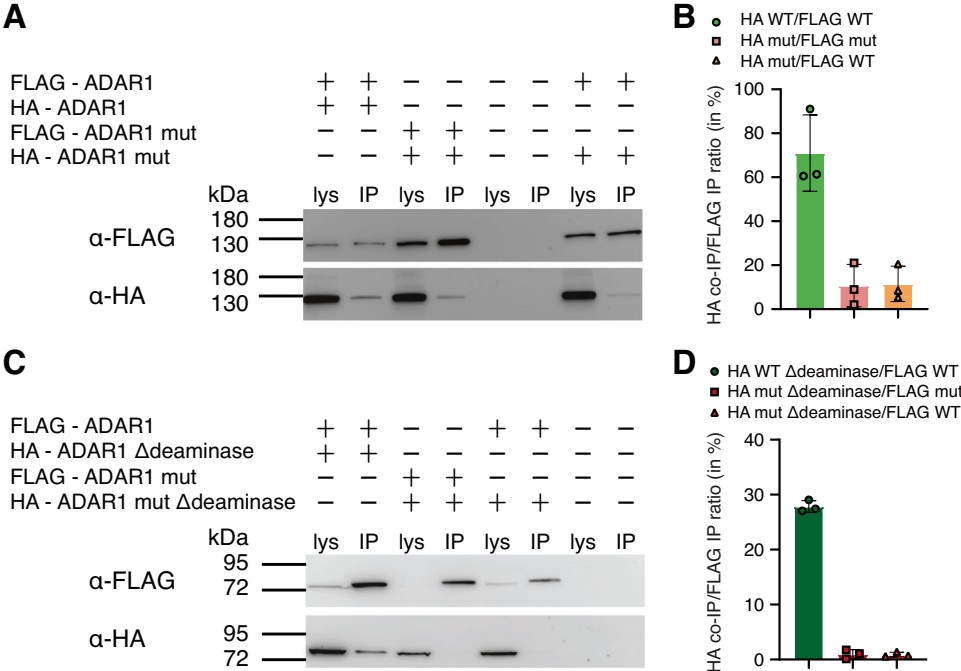

**Fig. 4 | ADAR1-dsRBD3 can mediate ADAR1 dimerization in vivo. A** Co-immunoprecipitation of FLAG-tagged and HA-tagged full length ADAR1 p110 (FLAG-ADAR1, HA-ADAR1). Both protein versions were transfected as indicated by + or − signs. After precipitating the FLAG-tagged version, the precipitate was tested for the presence of the HA-tagged protein in the presence of RNases. Mutations in dsRBD3 that prevent dimer formation (i.e. V747A, D748Q, W768V and C773S – FLAG/HA-ADAR1 mut) still allow dimer formation of full length ADAR1 p110, likely via the deaminase domain. **B** Quantification of full-length Co-IP/IP ratios of **A** and Supplementary Fig S11. Three independent blots were used to quantify individual bands of immunoprecipitated proteins. Data height correspond to mean values, and error bars indicate standard deviation (*n* = 3). **C** In the absence of the deaminase domain (Δdeaminase), the dimer forming surface on dsRBD3 becomes essential for dimer formation. FLAG-tagged full length ADAR1 (FLAG-ADAR1) interacts with HA-tagged ADAR1 without a deaminase domain (HA-ADAR1 Δdeaminase). However, when the dsRBD3 is mutated to prevent dimer formation (FLAG-ADAR1 mut) the interaction with a deaminase deficient ADAR1 is disrupted (HA-ADAR1 mut Δdeaminase). **D** Quantification of Co-IP/IP ratios of **C** and Supplementary Fig. S11. Three independent blots were used to quantify individual bands of immunoprecipitated proteins. Data height correspond to mean values, and error bars indicate standard deviation (*n* = 3). The ratio of HA-tagged/FLAG-tagged precipitated proteins were quantified using ImageJ (lys: lysate; IP: immunoprecipitation). Blots are probed with anti-FLAG (α-FLAG) and anti-HA (α-HA) antibodies. Source data are provided as a Source Data file.

p110 but also in the context of ADAR1 p150 (Fig. 5D, E), while site 6 showed strongly reduced editing for the ADAR1 p110 dsRBD3 mutant (Fig. 5E). Similar findings were observed for *Nicn1* (Supplementary Fig. S12). While the dimerization mutant showed reduced editing at some sites, other sites were not affected by the mutation. To test, whether this different behaviour might be the result of structural or geometric distributions of editing sites as suggested by Uzoniy et al.[52], we plotted editing sites that were not affected (green), mildly affected (orange), or strongly affected (red) by the dimerization mutation in different colours over a secondary structure prediction of the *Nicn1* 3' UTR (Fig. 5F). As can be seen from this analysis, no obvious structural or sequence constrain that would explain the observed differences can be identified. For instance, most sites, irrespective of their response towards the dimerization mutant are located opposite U residues and are in seemingly stable, double-stranded structures. Thus, in general we find that the dimerization mutant can reduce editing at specific sites, while not affecting editing levels at other sites.

## Discussion

Here we identify a dimerization interface in ADAR1, which is located at the level of the third dsRBD of this protein. We had shown earlier that ADAR1-dsRBD3 holds an N-terminal α-helical extension (α$_N$) that is important to position the N-terminal part of a bimodular nuclear localization signal that is recognized by Transportin-1 (Trn1)[7,53]. The crystal structure of this dsRBD could here independently confirm the N-terminal α$_N$-extension, originally identified by NMR. Interestingly,

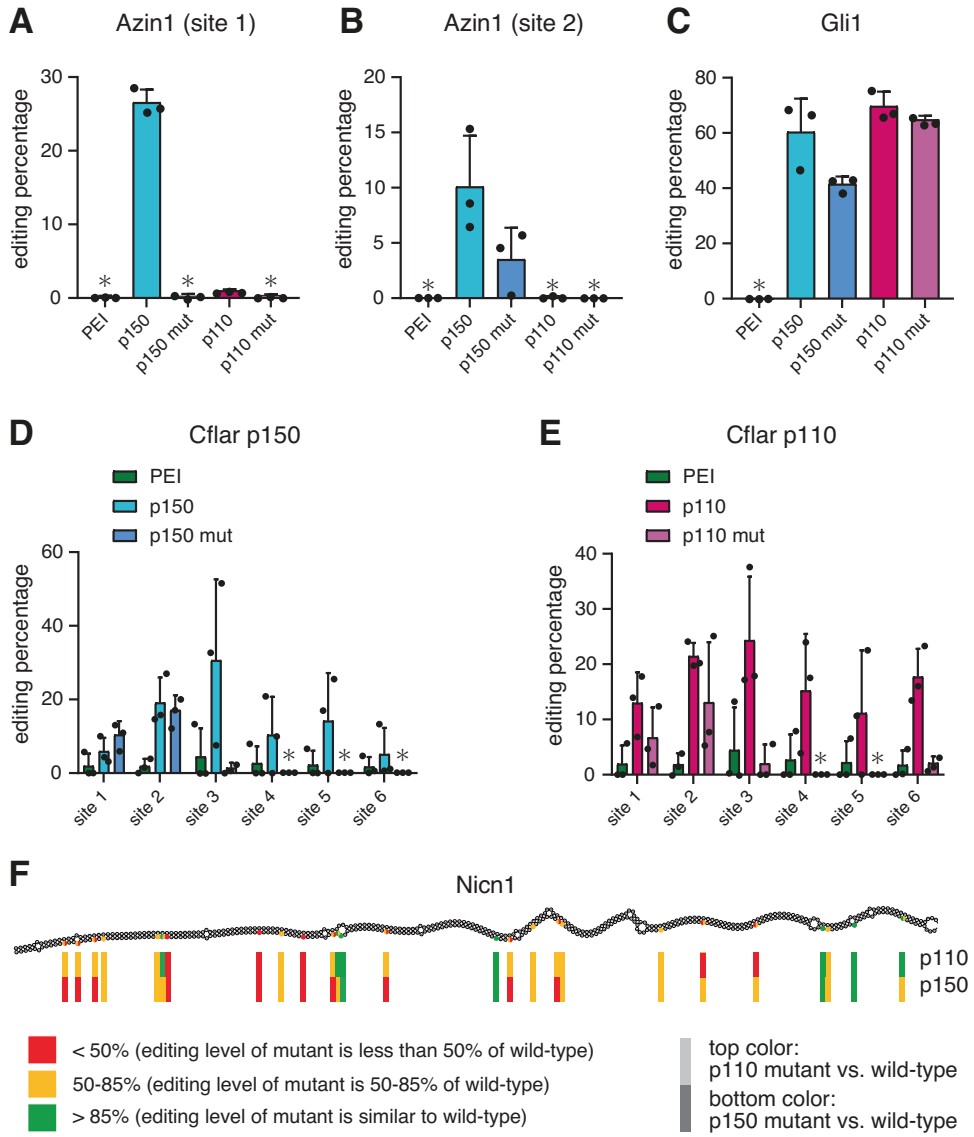

**Fig. 5 | dsRBD3 dimer interface mutations affect editing activity of ADAR1 in vivo.** HEK293T were transfected with full length ADAR1 p150 (p150), ADAR1 p110 (p110), ADAR1 p150 dsRBD3 mutant (p150 mut) and ADAR1 p110 dsRBD3 mutant (p110 mut). After transfection, total RNAs were extracted and different targets were amplified from cDNA. Amplicons were sequenced with Sanger sequencing and editing levels were determined by the relative height of G over A peaks. **A** *Azin1* site 1 is exclusively edited by ADAR1 p150 and a dimerization mutation abolishes editing. **B** Editing at site 2 in the same *Azin1* transcript is strongly reduced upon inhibition of dimer formation. **C** Editing of *Gli1* is reduced by the dimerization mutant but only in the context of ADAR1 p150. **D**, **E** Editing in the Alu element embedded within the *Cflar* transcript show that sites 1 and 2 are not affected by the loss of dimerization while editing of sites 3, 4, 5, and 6 is strongly reduced upon loss of

dsRBD3 dimerization. PEI serves as a negative control, in which cells were transfected with transfection reagent only. Asterisks (*) denote no measurable editing. Data height correspond to mean values, and error bars indicate standard deviation from 3 independent experiments (*n* = 3). **F** Graphical depiction of changes in editing observed in the dimerization mutants for ADAR1 p110 and the corresponding mutant isoform (top row) and ADAR1 p150 and the corresponding dimerization mutant (bottom row) in the editing sites of the 3'-UTR of *Nicn1*. Sites that are not affected by editing are depicted in green, sites that show less than 50% reduction in editing are highlighted in orange, while sites showing a reduction >50% of editing in the mutant are labelled in red. See also Supplementary Fig. S12. Source data are provided as a Source Data file.

however, the crystal structure of ADAR1-dsRBD3 revealed a dimerization interface, which seems unique for this particular domain as amino acids later identified to be crucial for dimerization (Supplementary Figs. S6 and S8) are not found conserved on other dsRBDs[35]. The insights gained from this study have important implications for our understanding of RNA editing and will be discussed below.

Dimerization of dsRBDs had already been noticed in the dsRBD-containing proteins PACT and TRBP[54], as well as Xlrbpa, the homologue of PACT in *Xenopus laevis*[55]. In these cases, dimerization is mediated by a Type B dsRBD, namely a dsRBD that lacks specific dsRNA-binding residues and therefore cannot bind to dsRNA[32]. The

homodimerization surface of these dsRBDs involve asymmetric contacts at the level of the third β-strand of the domains. The homodimer is indeed assembled through an inter-subunit parallel β-sheet where the two β3 strands interact via a register shift, thereby creating asymmetry[56]. In the case of ADAR1-dsRBD3, the dimerization reported here is clearly symmetric and consists of a back-to-back interaction of the two individual β-sheets (Fig. 1), which differs from the continuous β-sheet described above. As a consequence, the symmetric homodimer of ADAR1-dsRBD3 display one set of peaks in NMR spectra (Supplementary Fig. S7)[7], in contrast to PACT-dsRBD3 asymmetric homodimer that exhibits two sets of peaks[56]. Homodimerization

mediated by a dsRBD has also been reported for the Staufen protein, albeit with a distinct structural organization. Again, a Type B dsRBD lacking dsRNA-binding capacities was shown to form a symmetric homodimer through a domain-swap mechanism involving an N-terminal extension to the core dsRBD structure[57]. The ADAR1-dsRBD3 dimerization reported here appears unique, since the homo-dimerization property is incorporated within a Type A dsRBD that has retained dsRNA-binding properties (Fig. 3), while displaying protein-protein interaction capacity. This feature is encoded in the dsRBD fold itself, and not mediated by extensions attached to the dsRBD core domain, as reported previously for Type A dsRBDs[58]. Regarding the amino acid conservation of the key residues identified, dimerization of ADAR1-dsRBD3 is certainly conserved in mammals and most likely also conserved in all vertebrates from fish to mammals (Supplementary Fig. S13).

The importance of ADAR dimerization for its activity has been proposed in early studies and became generally accepted in the community. In the absence of structural data supporting dimerization, this assumption was, however, based on indirect observations and mutants that did not target dimerization per se, but dsRNA-binding or deaminase activity of one monomer in the complex[29,31]. Recently, dimerization properties of ADARs started to be unveiled with the description of an interaction interface between deaminase domains in ADAR2[27]. In the course of our experiments, we also noticed that the deaminase domain of ADAR1 can mediate self-association (Fig. 4A), which is in good agreement with the above-mentioned study[27]. Still, our mutational analysis proved that dimerization of ADAR1 can occur in the absence of the deaminase domain in a dsRBD3-dependent manner (Fig. 4C, D). Importantly, for the sake of comparison, interaction via the deaminase domains is not strong enough to mediate dimerization in vitro in the absence of dsRNA[27], whereas interaction via dsRBD3, as reported here, is strong enough to mediate dimerization even in the absence of dsRNA, and only dimeric forms of the constructs are observed in vitro (Fig. 2 and Supplementary Fig. S1). Several studies carried out on ADAR1 and/or ADAR2 pointed for an interface of interaction between the monomers to occur over a widespread region including the deaminase domain as well as the dsRNA-binding domains[29,31,33,59]. Our data, together with previous reports, therefore suggest that dsRBD3 is the main driver of dimerization in ADAR1 (Fig. 4B), whereas dsRBD1 would be the equivalent region in ADAR2[59]. On top of that, another layer of dimerization is provided by the deaminase domains, the association of which may be enhanced in the presence of dsRNA[27]. Heterodimerization of ADAR1 and ADAR2 has been observed by in vivo FRET experiments, and proposed as an interesting mechanism to regulate efficiency and specificity of editing[30]. Considering the absence of dsRBD3 in ADAR2, and the poor conservation of dsRBD1 between ADAR1 and ADAR2[9], association of ADAR1 with ADAR2 is likely to occur via heterodimerization of their deaminase domains, whose interaction surfaces are conserved[27]. This suggests that assemblies composed of ADAR1 and ADAR2, if they exist at all, would be dimers of dimers bridged via their deaminase domains, an intriguing architecture that would deserve to be examined structurally.

Although only dimeric forms of ADAR1-dsRBD3 were observed in vitro in our study, it would be interesting to know whether the ADAR1-dsRBD3 dimer can dissociate to form monomers under certain conditions. For example, it has been reported that the Dicer/ADAR1 complex is likely to consist of one Dicer and one ADAR1 molecule, but not an ADAR1 homodimer[33]. The ADAR1 homodimer could therefore dissociate upon binding to other protein partners. This possibility is particularly relevant in the case of ADAR1-dsRBD3 interaction with the nuclear import receptor Trn1. In fact, we previously constructed a model of the interaction between Trn1 and ADAR1-dsRBD3 as a monomer, since its dimerization capabilities were unknown at the time[7]. Although it might seem structurally feasible for ADAR1-dsRBD3 to bind to Trn1 as a dimer, since the interaction interface is basically opposite to the dimerization interface, the structural model we previously constructed is not directly compatible with the binding of a dimer of ADAR1-dsRBD3 to Trn1. There are indeed several clashes between the second ADAR1-dsRBD3 monomer and Trn1 in this overall organisation. From a structural point of view, ADAR1-dsRBD3 should either bind to Trn1 with a different orientation, allowing a second dsRBD to fit at the exit of the Trn1 concave pocket, or bind as a monomer, with dissociation of the dimer that could be favoured upon binding to Trn1. Further studies are required to distinguish between these possibilities.

Importantly, although dsRBD3 interface mutants retain dsRNA-binding properties, the mutations affecting dimerization have various impacts on RNA editing. For instance, the ADAR1 p150 dsRBD3 interface mutant is unable to target site 1 in *Azin1*. Similarly, dsRBD3 interface mutants of both ADAR1 p150 and p110 show altered editing patterns in the SINE-containing RNAs *Cflar* and *Nicn1*. However, editing of *Gli1* is unaffected in ADAR1 p110 dsRBD3 interface mutant and only slightly affected by the mutations in ADAR1 p150. Of note, mixed effects have also been observed with mutants affecting dimerization at the level of the deaminase domain in ADAR2, with editing sites that are either affected or insensitive to these mutations[27]. The case of *Nicn1* nicely demonstrates that even nearby sites can be differently affected by the dimerization mutation (Fig. 5F). Currently, no obvious rules such as opposing base, or structural context can explain why only some sites are affected by the lack of dimerization. Clearly, additional studies will be required to determine how dimerization may affect editing of specific substrates.

We showed that dsRBD3 can bind dsRNA as a dimer (Fig. 3 and Supplementary Fig. S9). Given the flexible linkers between the dsRBDs of ADARs, but also between the dsRBD3 and the deaminase domain, it is difficult to predict how a substrate would be bound. While it is possible that the dsRBDs of one ADAR interact with one substrate, while the dsRBDs of the dimerization partner bind to another substrate, it is perfectly possible that the dsRBDs of both dimerized ADARs bind to the same substrate. In such a case, the region between the binding sites of one ADAR and that of the dimerization partner would need to be looped out. Double-stranded RNA is rather rigid making a looping less likely. However, single stranded RNA is very flexible and could therefore, within a few nucleotides, allow complete flexibility of two distinct double-stranded regions, such as adjacent stem-loops in the same RNA, which could be clamped by the dsRBD3 dimer.

Interestingly, our finding, that some but not all editing sites are affected when the dimerization surface of dsRBD3 is mutated suggests that dimerization is only required for efficient editing of some sites. Most importantly, this shows that dimerization has different impact on different sites, and as such, inhibition of dimerization, as opposed to inhibition of catalytic activity, might be an attractive way to pursue for modulating ADAR1 editing activity in the context of immunotherapy[26,60,61]. Towards this goal, deciphering the substrates affected by dsRBD3 dimerization will require more in-depth studies on how dimerization mutations affect transcriptome-wide editing. Positioning of affected editing sites relative to each other, would clearly shed some light on the architecture of the substrate-ADAR complexes. In this context it is interesting to note that a periodicity of editing sites has already been proposed to be affected by the binding mode of ADARs to their substrates[52,62]. In combination with the determination of the structure of full-length ADAR1 in complex with RNA substrates, such data would certainly provide some answers to the puzzling questions regarding the mechanisms of editing-site selection by ADARs.

## Methods

### Cloning, expression, and protein purification for structural studies

The DNA sequence encoding the third dsRBD of human ADAR1 (residues 688-817: dsRBD3-long; residues 708–801: dsRBD3-mid; residues

716–797: dsRBD3-short) (Uniprot entry P55265) were subcloned by PCR amplification from full-length ADAR1, between *BamHI* and *XhoI* cloning sites in a modified pET28a expression vector containing an N-terminal tag His$_6$-tag and a TEV protease cleavage site. These constructs with cleavable His$_6$-tag were used in initial SEC-SAXS experiments (i.e. dsRBD3-long, dsRBD-mid, and dsRBD3-short) and in crystallization and structure determination (dsRBD3-short). Residues corresponding to human ADAR1-dsRDB3 (residues 708–801) were also cloned in a pET28a vector between *NdeI* and *XhoI* cloning sites. This construct with uncleavable His$_6$-tag (i.e. ADAR1-dsRBD3), and the associated mutants generated by PCR amplifications[63] (i.e. ADAR1-dsRBD3 V747A + D748Q + W768V + C773S, and chimeric ADAR1-ds3/XIrbpa-ds2), were used in SEC-MALLS, SEC-SAXS, NMR and ITC experiments. All proteins constructs were overexpressed in *E. coli* BL21(DE3) Codon-plus (RIL) cells in either LB media or M9 minimal media supplemented with $^{15}NH_4Cl$. The cells were grown at 37 °C to OD600 ~ 0.5, cooled down at 30 °C, and induced at OD600 ~ 0.6–0.7 by adding isopropyl-β-D-thiogalactopyranoside to a final concentration of 1 mM. Cells were harvested 16–20 h after induction by centrifugation. Cell pellets were resuspended in lysis buffer [50 mM Tris-HCl (pH 8.0), 1 M NaCl, 20 mM imidazole, 1 mM DTT, 1 mM EDTA] and lysed by sonication at 4 °C. Supernatant was loaded on an Ni-NTA column on an ÄKTA purification system (Cytiva), and the protein of interest was eluted with an imidazole gradient. The fractions containing the protein were pooled and the His$_6$-tag was either cleaved off by an overnight incubation with TEV protease at 20 °C and removed by a second passage onto the Ni-NTA column, or kept in the final purified constructs. The proteins were further purified by size-exclusion chromatography (Superdex 75 HiLoad 26/600, Cytiva) in the storage buffer [50 mM Tris-HCl (pH 8.0), 100 mM KCl, 1 mM TCEP], concentrated to ~15–20 mg/mL using Amicon 3000 MWCO (Millipore), and kept at −20 °C until further use in crystallization or biophysical assays. For specific experiments, proteins were dialysed against the SEC-SAXS buffers [20 mM Na-HEPES pH 7.3, 55 mM KOAc, 10 mM NaCl, 1 mM TCEP] or [20 mM Na-phosphate pH 7.0, 100 mM NaCl, 2 mM 2-mercaptoethanol] or the ITC buffer [25 mM Na-phosphate pH 7.0, 100 mM NaCl, 2 mM 2-mercaptoethanol].

## SAXS data collection and analysis

SEC-SAXS experiments were performed at the SWING beamline at the SOLEIL synchrotron (Saint-Aubin, France) using an online high-performance liquid chromatography (HPLC) system equilibrated at 15 °C and at a flow rate of 0.3 mL/min. All scattering intensities were collected on the elution peaks after injection on a BioSEC-3 column (Agilent) equilibrated in 20 mM Na-HEPES pH 7.3, 55 mM KOAc, 10 mM NaCl, 1 mM TCEP for the initial ADAR1-dsRBD3 constructs, or in 20 mM Na-phosphate pH 7.0, 100 mM NaCl, 2 mM 2-mercaptoethanol for the dimerization mutants. Data were processed using FOXTROT[64] and further analysed using the ATSAS 3.0 suite of programs[65]. Data were inspected using SHANUM[66] and cropped using DATCROP[65]. The radius of gyration (Rg), the maximum dimension Dmax, as well as the molecular weight were calculated using default parameters in PRIMUS[67]. Normalized Kratky plots were prepared using BioXTAS RAW[68]. CRYSOL[69] was used to calculate theoretical scattering curves from the crystal structure atomic coordinates of dimeric and monomeric ADAR1-dsRBD3 (dsRBD-short construct). Missing side chains in the models were previously modelled in COOT assuming the most prevalent rotamer conformation. Ab initio shape reconstruction was also performed from the pairwise distance distribution function plot generated from GNOM using the DAMMIN and GASBOR programs[70,71]. Ten independent model building runs were performed in each case, while imposing a 2-fold symmetry condition, and the ab initio models were aligned and analysed using DAMAVER[72]. Averaged bead models were also computed for the most populated cluster of each ensemble and

aligned with the crystal structure of dimeric ADAR1-dsRBD3 using CIFSUP.

## Crystallization, data collection, structure determination and refinement

Crystals of ADAR1-dsRBD3 (dsRBD-short construct, residues 716–797) were obtained at 5–8 mg/mL of protein by vapour diffusion against 100 mM sodium citrate pH 4.0, 20% (w/v) PEG 6000, and 1.0 M LiCl. Crystals were cryoprotected using 18–20% (v/v) glycerol before flash freezing. Data were collected on a single crystal at the micro focused PX2 beamline at the SOLEIL synchrotron (Saint-Aubin, France). Diffraction data were indexed, processed, merged and scaled using XDS[73] and AIMLESS[74]. Phases were determined by molecular replacement using PHASER[75] in the CCP4 Suite of programs[76] and the first structure of the NMR ensemble (PDB 2MDR) as a model. Model building of ADAR1-dsRBD3 was first performed with ARP/wARP using the classic warpNtrace autotracing[77]. Restrained refinements were then carried out up to 1.65 Å with the program REFMAC[78], COOT[79] and Phenix[80]. Atomic displacement parameters were modelled using a hybrid TLS + B$_{iso}$ model, using one TLS group for each protein monomer[81].

Crystals of ADAR1-dsRBD3/dsRNA complex were obtained by mixing one equivalent of protein with 2 equivalents of RNA in 50 mM Tris-HCl pH 8.0, 100 mM KCl and 1 mM TCEP and left on ice for 1 h. The RNA sequence of 13-nucleotides employed here (i.e. 5′-CGAAGC-CUUCGCG-3′) has the capacity to base-pair with itself to form short double-stranded RNA modules. In addition, the nucleotide overhangs allow these RNA modules to self-assemble into a long RNA-helix, thus promoting crystal packing. Crystals of freshly prepared complex were obtained at 7 mg/mL of protein by vapour diffusion against 50 mM sodium cacodylate pH 6.0, 5% (w/v) PEG 4000, 30 mM CaCl$_2$, and 230 mM KCl. Crystals were cryoprotected using 20% (v/v) glycerol before flash freezing. Data were collected on a single crystal at the micro focused PX2 beamline at the SOLEIL synchrotron (Saint-Aubin, France). Diffraction data were indexed and integrated using XDS[73] and processed using STARANISO[82] in order to take into account the moderate anisotropy of the dataset, that diffracted up to ~2.8 Å along the best direction, but only up to ~3.3 Å in the weakly diffracting ones. Phases were determined by molecular replacement using PHASER[75], and different combinations of models consisting of ADAR1-dsRBD3 monomer structure, double-stranded 13-nucleotides A-form RNA generated in COOT, and single-stranded 13-nucleotides adopting an A-form-like RNA structure. A complete solution (TZF = 39.0) without any missing entity in the asymmetric unit was found by searching two dsRBDs, one dsRNA module (2 ×13-nucleotides), and one ssRNA molecule (13-nucleotides). Restrained refinements were carried out up to 2.8 Å with the program LORESTR[83], REFMAC[78], and Phenix[80]. Model and map visualizations for manual reconstruction were performed with the program COOT[79]. Atomic displacement parameters were modelled using a pure TLS model, using one TLS group for each protein or RNA chain[81].

## SEC-MALLS

Size-exclusion chromatography in line with multi-angle laser light scattering (SEC-MALLS) experiments for absolute mass determination were conducted at 25 °C with a BioSEC-3 column (Agilent) equilibrated in 20 mM Na-phosphate pH 7.0, 100 mM NaCl, 2 mM 2-mercaptoethanol, on a high-performance liquid chromatography (HPLC) system (Prominence, Shimadzu) at a flow rate of 0.4 mL/min. The HPLC system is coupled to an UV detector (SPD-20, Shimadzu), an Optilab® T-rEX™ differential refractometer (Wyatt Technology) and a miniDawnTM TREOS Multi Angle Laser Light Scattering (MALLS) detector (Wyatt Technology). All data were collected and analysed with ASTRA software (Wyatt Technology).

## NMR spectroscopy

All NMR spectra were measured at 35 °C on Bruker AVIII-HD 700 MHz spectrometer equipped with a TCI 5-mm cryoprobe using 5-mm Shigemi tubes. The data were processed using Topspin 3.6 (Bruker) and analysed with NMRFAM-SPARKY[84]. ADAR1-dsRBD3 mutant and chimeric proteins were checked to be properly folded by running ($^{15}$N,$^1$H)-HSQC spectra in the NMR buffer (20 mM Na-phosphate pH 7.0, 100 mM NaCl, 2 mM 2-mercaptoethanol).

## Isothermal titration calorimetry

To assess the nucleic acid binding properties of ADAR1-dsRBD3 interface mutant, we produced RNA substrates by in vitro transcription with T7 polymerase, namely a dsRNA duplex of 24 bp of arbitrary sequence (RNAfwd of sequence 5′-GGGAUCAAUAUGCUAAGCG AUCCC-3′ and RNArev being the reverse complement)[46]. DNA templates (i.e. 5′-GGGATCGCTTAGCATATTGATCCCTATAGTGAGTCGTAT TA-3′ and 5′-GGGATCAATATGCTAAGCGATCCCTATAGTGAGTCGTA TTA-3′), as well as the top strand T7 promotor primer (i.e. 5′-TAA TACGACTCACTATAG-3′), were purchased from Eurogentec. RNAs were purified by anion-exchange chromatography (Mono Q 10/100 GL, Cytiva) under native conditions (25 mM Na-phosphate pH 6.5, 50 mM NaCl, 5 mM MgSO$_4$)[85]. ITC experiments were performed on a VP-ITC instrument (MicroCal) calibrated according to the manufacturer's instructions. The samples of protein and nucleic acids were prepared in and dialysed against ITC buffer (25 mM Na-phosphate pH 7.0, 100 mM NaCl, 2 mM 2-mercaptoethanol). The concentration of protein and nucleic acid was determined using OD absorbance at 280 and 260 nm, respectively. The sample cell (1.4 mL) was loaded with 2.25 μM of the dsRNA duplex and the concentration of ADAR1-dsRBD3 wild type and variants in the syringe was 80 μM. Titration experiments were done at 25 °C with a stirring rate of 307 r.p.m. and consisted of 34 rounds of 8-μL injections. Data were plotted and analysed using MicroCal PEAQ-ITC analysis software v1.1, using equations for a single binding-site model.

## Plasmids for HEK293T and HeLa transfections

Human ADAR1 plasmids were made as described in our previous studies[50]. All cloning and mutation procedures were done by NEBuilder HiFi DNA assembly according to the manufacturer's instructions. All the plasmids and the oligos used for cloning and transfections are listed in Supplementary Tables S3 and S4. Western blots with FLAG or HA-tag antibodies were used for monitoring ADAR1 expression.

## Cell cultures and transfections

Henrietta Lacks (HeLa, ATCC CCL-2™) and Human Embryonic Kidney 293T (HEK293T, ATCC CRL-3216™) cells were cultured in high-glucose Dulbecco's modified Eagle's medium (DMEM) (Thermo Fischer Scientific, Waltham, MA) supplied with 10% fetal bovine serum, pyruvate and L-glutamine. Cells were incubated at 37 °C with 5% CO$_2$ saturation.

Polyethyleneimine (PEI) transfections were done according to polyplus jet-PEI$^R$ manufacturer's instructions. HeLa cells were used for immunostainings and co-immunoprecipitation. HEK293T cells were used for in vivo editing assays.

## Co-immunoprecipitation

HEK293T cells were transfected with either FLAG-tagged or HA-tagged ADAR1 p110 wild-type or mutant. Cells were lysed in the IP buffer [25 mM Tris-HCl pH 7.4, 150 mM NaCl, 1 mM EDTA, 1% NP-40, 5% glycerol and 1x protease inhibitor (Roche)]. The samples were lysed by BIORUPTOR® COOLER (diagenode, Belgium) using a protocol of 8 cycles of 40 s *on* and 60 s *off*, and were then treated with RNase A. The magnetic FLAG M2 beads (M8823 Merck Millipore, USA) were equilibrated in the lysis buffer and blocked with 5% BSA and 5% dry milk powder. The clarified clear lysate was incubated with the beads for 2 h at 4 °C on a rotating wheel. After that the beads were washed with IP buffer 5 times. The bound proteins were eluted with 2xSDS loading dye at 95 °C for 5 min. The samples were then analysed by western blotting.

## Western blot, quantification and immunofluorescence

Ectopic expression of ADAR1 was monitored by western blotting and immunofluorescence. The cell pellet was lysed in 2xSDS sample buffer and sonicated. Lysates were separated on 8% SDS-PAGEs and transferred to Nitrocellulose membranes using a Wet-transfer. The Membranes were blocked in 1× TBST supplemented with 5% dry milk powder. ADAR constructs were detected with an anti-FLAG rabbit polyclonal antibody (Sigma Aldrich, Cat: F7425, Size: 1000 μL, Conc: 2 mg/mL, Lot: #252651, St. Louis, MI), directed against the N-terminus, or an anti-HA.11 Epitope rabbit antibody (BioLegend$^R$, Cat: 902301, Size: 200 μL, Conc: 1 mg/mL, Lot: B279476, St Diego, CA). The antibodies were diluted at &:1000 in 1xTBST with 3% dry milk powder as per the manufacturer's instructions. After incubation with the primary antiserum the blots were detected with a peroxidase-conjugated goat anti rabbit or mouse secondary antibody using a BioRad Chemoluminescent detection kit. The specificities of the primary antibodies were tested by untransfected controls and the validation report for the antibodies are available on the supplier's websites.

The density of immunoprecipitated bands of FLAG and HA were calculated by FIJI, an extended version of ImageJ[86]. The density of co-precipitated HA-bands was divided by FLAG-bands to get the ratios. The data represents average of three independent experiments. The bar graphs of the quantification were generated by Prism.

Subcellular localization of ADAR1 variants were confirmed by immunofluorescent staining. In short, cells were washed with 1x PBS and fixed on coverslips using a cell-fixation solution and methanol. Then, the localization was made visible by immunofluorescence staining using an anti-FLAG rabbit polyclonal antibody (Sigma Aldrich, Cat: F7425, Size: 1000 μL, Conc: 2 mg/mL, Lot: #252651, St. Louis, MI) in combination with Alexa 546 secondary antibody (Invitrogen, Thermo Fisher Scientific, Waltham, MA) and the nuclei was counterstained with DAPI. Microscopic confocal sections were taken on Confocal Laser Scanning Microscope FV3000 (Olympus, Tokyo, Japan).

## RNA isolation and editing assays

RNA isolation from the HEK293T cells was done using TRIzol$^{TM}$ reagent (Thermo Fischer Scientific, Waltham, MA) according to manufacturer's instructions. Isolated RNA was treated with DNaseI (New England Biolabs, Ipswich, Massachusetts) and subsequently purified by phenol:chloroform, chloroform extraction and precipitated with ethanol. The RNA was reverse transcribed using LunaScript reverse transcriptase (New England Biolabs, Ipswich, MA), random hexamer primers, Oligo(dT)s and RNase inhibitor murine. The cDNA was then used for PCR with the OneTaq 2x Master mix with Standard Buffer (New England Biolabs, Ipswich, Massachusetts). Targets for differential editing of the mutant in p150 and p110 were: *Azin1*, *Gli1* and *Cflar*. The amplicons were resolved on 2% Agarose gel to verify the size. Then a gel elution kit was used, and the samples were sent for Sanger sequencing (Eurofins, Luxembourg). The chromatograms were aligned with Geneious (Version 11.1.5) and the editing percentage was calculated with the SnapGene Viewer software. The genomic coordinates for all editing sites are provided in Supplementary Table S5.

## Reporting summary

Further information on research design is available in the Nature Portfolio Reporting Summary linked to this article.

# Data availability

The crystal structures of ADAR1-dsRBD3 dimer free and bound to dsRNA have been deposited to the Protein Data Bank (PDB) under accession codes 7ZJ1 and 7ZLQ, respectively. We uploaded the SAXS

data to the Small-Angle X-ray Scattering Biological Data Bank (SASBDB)[87] with the following accession codes: SASDVF7 (dsRBD3-long), SASDVG7 (dsRBD3-mid), SASDVH7 (dsRBD3-short), SASDVJ7 (ADAR1-dsRBD3), SASDVK7 (interface mutant), SASDVL7 (chimeric ADAR1-ds3/XIrbpa-ds2). Further, the cited structures in this paper can be found with the following accession codes: 2MDR (NMR structure of ADAR1-dsRBD3 monomer); 1DI2 (structure of XIrbpa-dsRBD2). The data supporting the findings of this study are available from the corresponding authors upon request. Source data for the figures and Supplementary figures are provided as a Source Data file. Source data are provided with this paper.

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

## Acknowledgements

We acknowledge SOLEIL for provision of synchrotron radiation facilities and we would like to thank William Shepard, Martin Savko and Serena Sirigu for assistance in using beamline PX2. We also thank Pierre Legrand for assistance in preliminary data collection on PX1, and for the development of the XDSME package. We are immensely indebted to Aurélien Thureau for assistance in SEC-SAXS data collection at the SWING beamline. We also acknowledge Franck Brachet for the access to the crystallography platform of the IBPC, Alexandre Pozza and Françoise Bonneté (IBPC) for assistance in SEC-MALLS data measurements and analysis, and Sylviane Hoos and Patrick England (Molecular Biophysics Facility – PFBMI, Institut Pasteur) for assistance in ITC data measurements. We also thank Alwine Hildebrandt and Tanja Rohr for excellent technical assistance. Access to the biomolecular NMR platform of the IBPC, supported by the CNRS, the Labex DYNAMO (ANR-11-LABX-0011), the Equipex CACSICE (ANR-11-EQPX-0008) and the Conseil Régional d'Île-de-France (SESAME grant) is acknowledged. P.B. and M.F.J. are supported by a joint ANR-FWF Grant (Nos. ANR-16-CE91-0003 to P.B. and FWF-I2893 to M.F.J.). Part of this work was also supported by grant F80-07 by the Austrian Science Foundation (M.F.J.). This research was funded in part by the Austrian Science Fund (FWF) [grant DOI: 10.55776/F80 and 10.55776/I2893]. For open access purposes, the author has applied a CC BY public copyright license to any author accepted manuscript version arising from this submission.

## Author contributions

P.B. and M.F.J. conceived the study and supervised the project. A.M. purified constructs for crystallization studies with support from M.C. and with input from P.B. and C.T.; A.M. and P.B. purified constructs for SEC-MALLS and SEC-SAXS measurements; A.M. set-up crystallization plates and collected diffraction data; A.M. and P.B. performed diffraction data processing and refinement, SEC-MALLS and SEC-SAXS data collection and analysis; P.B. purified constructs, measured and analysed NMR and ITC data. V.R., S.M. and T.C.M. performed in vivo experiments and editing assays with input from M.F.J. M.F.J. and P.B. wrote the manuscript with input from A.M. and V.R.

## Competing interests

The authors declare no competing interests.
