## [Transparent Peer Review file · Nature Communications]

Dimerization of ADAR1 modulates site-specificity of RNA editing

Corresponding Author: Dr Pierre Barraud

Version 0:

Reviewer comments:

Reviewer #1

(Remarks to the Author)

In the submitted manuscript, the authors reported their analysis of the third dsRNA-binding domain of human ADAR1. Their structural analysis revealed an extended dimerization interface mediated by β -sheets that is distinct from the region required for RNA-binding. They also generated a quadruple mutant (V747A, D748Q, W768V and C773S) that is defective in dimerization and showed that this mutant exhibits an impaired ability in editing of selected sites. Overall, I find the work interesting and think that it will be useful to the RNA community. Below are some comments that might improve the manuscript further.

Major comments:

- 1) The authors transfected "FLAG-tagged ADAR1 p110 and ADAR1 p150 wildtype or a version harboring the dimerization mutations in dsRBD3 into HEK293 cells" and found no change in localization. Since the deaminase domain can also mediate dimerization, can the authors also check constructs with the deaminase domain deleted? (This is of course assuming that the deaminase domain does not play a role in localization.)
- 2) Figure 5: Were the editing level measurements carried out in wildtype or ADAR1 KO (without endogenous ADAR1) cells? If wildtype cells were used, this might explain why GLI1 did not show a substantial reduction in editing.
- 3) Instead of selected sites, the authors can consider performing transcriptome-wide RNA-seq analysis to observe global changes in editing. What are the differences between sites that are unaffected and sites that show significantly reduced editing?
- 4) I am curious to know if this dsRBD3-mediated dimerization is conserved between different primates or mammals. Are the key residues conserved across various primates or mammals?

Minor comments:

- 5) How did the authors decide on the boundaries of dsRBD3-long (residues 688-817) and dsRBD3-mid (residues 708-801)? Why 688-817 and 708-801?
- 6) Acronyms should be spelled out in full on first use.

Reviewer #2

(Remarks to the Author)

ADAR1 is an important enzyme in vertebrates that deaminates adenine bases to inosine in the context of dsRNA, and plays important roles in post-transcriptional RNA modification and innate cellular immunity. Mboukou et al present a timely structural and functional study of dimerization in ADAR1 by examining the homodimerization of the third dsRNA binding domain of this protein (dsRBD3), without and with dsRNA. This brings new data that weren't accessible in prior NMR studies of this domain and sheds a new perspective on the architecture of RNA editing proteins. This has important implications for understanding their mechanism in a cellular context.

By combining solution data (SAXS), crystal structures, co-immunoprecipitation and editing assays, the authors provide an excellent body of data from well-executed experiments to support their new insight that ADAR1 dsRBD3 forms a

symmetrical homodimer. Moreover, they show, with a crystal structure, that this homodimer can bind two molecules of dsRNA in a back-to-back orientation. By differential tag co-immunoprecipitations, the authors show that mutations or deletions that impact dimerization in vitro can affect co-precipitation from cells. Finally, they demonstrate that loss of dimerization of dsRBD3 affects a subset of editing sites.

This is an important piece of work and should be published, albeit with some modifications to improve data presentation/interpretation and clarity for the reader.

1. Introduction, paragraph 2. The phenotypic data that are discussed relate to mice – this should be made clear in the text.
2. Results paragraph 2: the buried surface area of the interface is denoted as 490 Å². The authors should clarify if this is the total buried surface area or the area per subunit.
3. The CRYSOLOG calculations show very convincingly that the dimer structure better represents the data than a monomer. It would be great if the authors could complement this by showing the fit of the crystal structure to a bead model of the dimer calculated e.g. by GASBOR.
4. The SAXS experiments show some evidence of aggregation and the SEC-SAXS profiles are also not symmetrical, indicating non-ideal behavior. The range of images used for SAXS curve calculation from the SEC-SAXS experiments should be indicated on the chromatograms and the SAXS scattering profiles derived from these images should also be shown in the supplement.
5. It would be easier to compare the Kratky plots in the supplementary data if the authors plotted as normalised Kratky.
6. It's rather surprising that the SEC-MALLS data show increasing Mw at later elution volumes. These SEC profiles are also non-symmetric. This suggests that there might be some non-ideal behavior, with the aggregated protein perhaps interacting with the matrix of the column. The limitations of these data should be discussed more fully.
7. Where the authors have used ITC data to show RNA binding in the dimeric dsRBD3 versus monomerising mutants, the difference in binding is quoted as "(factor 3.3 +/- 0.8)". It's not really clear how +/-0.8 is calculated for a factor or what it really means. I think it would be clearer to simply quote the Kd value of the WT ADAR3 dsRBD3 RNA binding here.
8. The co-IP data are really important to the manuscript but this is the least well-executed component. It is not clear how many different experiments have been undertaken; this should be stated clearly. They should also clarify in the main body of the text that these experiments are done with full length proteins and use more informative labelling on in the figure i.e. ADAR1, rather than "A1".
9. The co-IPs also show some variation in the transfection efficiency (annoying, but a common issue) as well as the efficiency of the pull-down (not ideal). This makes it a bit difficult to compare across samples in these experiments. Can the authors provide some quantification of the co-IPed HA-tagged protein bands versus the IPed FLAG-tagged protein bands? This would make the differences between the ADAR1 dsRBD3 mutations versus the delta-deaminase constructs clearer.
10. The editing assays are a further important validation of the functional importance of the dsRBD3 dimerization. There seems to be quite a bit of variation depending on whether sites are mostly edited by one isoform or another of ADAR1. The discussion could be briefly expanded to highlight that data for some sites is noisier than others.
11. The new structure of ADAR1 dsRBD3 alters our understanding of prior models of dsRBD3 contributing to nuclear import via interactions with transportin. Given that previous models were based on limited data and that there are now better modelling tools available, could the authors re-evaluate/regenerate the model with transportin they presented in Barraud et al 2014?

Version 1:

Reviewer comments:

Reviewer #1

(Remarks to the Author)

The authors submitted a revised manuscript on their study of ADAR1 dimerization. They have addressed my previous comments satisfactorily and I recommend publication in Nature Communications.

Reviewer #2

(Remarks to the Author)

I am satisfied that the authors have made every effort to address all comments that I raised on review. Consequently, the SAXS and IP data, as now presented, are robust and support the authors' conclusions. This augments the other structural data that are presented in the manuscript that are of a very high standard. The work is a thorough analysis and worthy of publication. It will be of great interest to the field.

Manuscript: NCOMMS-23-59848

Manuscript Title: Dimerization of ADAR1 modulates site-specificity of RNA editing

Allegra Mboukou, Vinod Rajendra, Serafina Messmer, Therese C. Mandl, Marjorie Catala, Carine Tisé, Michael F. Jantsch & Pierre Barraud

Response to the reviewers' comments

We thank the three reviewers for their positive and constructive comments. We are happy to submit a revised version of our manuscript and hope that it is now suitable for publication in *Nature Communications*. Below we address each of the reviewers' comments:

Reviewer #1

In the submitted manuscript, the authors reported their analysis of the third dsRNA-binding domain of human ADAR1. Their structural analysis revealed an extended dimerization interface mediated by β -sheets that is distinct from the region required for RNA-binding. They also generated a quadruple mutant (V747A, D748Q, W768V and C773S) that is defective in dimerization and showed that this mutant exhibits an impaired ability in editing of selected sites. Overall, I find the work interesting and think that it will be useful to the RNA community. Below are some comments that might improve the manuscript further.

We appreciate the positive feedback on our work.

Major comments:

1) The authors transfected "FLAG-tagged ADAR1 p110 and ADAR1 p150 wildtype or a version harboring the dimerization mutations in dsRBD3 into HEK293 cells" and found no change in localization. Since the deaminase domain can also mediate dimerization, can the authors also check constructs with the deaminase domain deleted? (This is of course assuming that the deaminase domain does not play a role in localization.)

We have followed this reviewer's suggestion and performed additional transfections, followed by antibody staining (Supplementary Figure S9A, the last two rows to the right side). We confirm that the deletion of deaminase domains in both the wild-type and dsRBD3 interface mutant does not alter cellular localization.

2) Figure 5: Were the editing level measurements carried out in wildtype or ADAR1 KO (without endogenous ADAR1) cells? If wildtype cells were used, this might explain why GLI1 did not show a substantial reduction in editing.

We believe that there is a misunderstanding of the experimental setup: We used wild-type HEK293T cells to study editing levels of different endogenous substrates. Endogenous Gli1 does not show background editing in these cells (see PEI-mock transfection). However, we see little difference in editing between the co-transfected wild type versions and dimerization mutant for either p150 or p110, indicating, that the Gli1 editing site does not require ADAR1 to dimerize.

3) Instead of selected sites, the authors can consider performing transcriptome-wide RNA-seq analysis to observe global changes in editing. What are the differences between sites that are unaffected and sites that show significantly reduced editing?

We thank the reviewer for this suggestion. Indeed, we have tested a few additional substrates. We are now also presenting data for the heavily edited 3'-UTR of Nicn1 where we have tested for

changes in editing at 26 closely spaced sites (Figure 5F and Supplementary Figure S11). In Figure 5F we have now highlighted sites on a folding prediction of the RNA that are not affected by the mutation (*green*), that are mildly affected (*orange*) and strongly affected (*red*). We have shown such differences for both the p150 and p110 versions. However, there seems no apparent logic as to whether a site is sensitive to the dimerization mutation or not. The differences are also shown in a bar graph in Supplementary Figure S11. Clearly, in the future, additional experiments will be required to understand the contribution of the dimerization at dsRBD3 to editing efficiency. We have now added the following statement to the results: “*Similar findings were observed for Nicn1 (Supplementary Figure S11). While the dimerization mutant showed reduced editing at some sites, other sites were not affected by the mutation. To test, whether this different behaviour might be the result of structural or geometric distributions of editing sites as suggested by Uzoniy et al. [52], we plotted editing sites that were not affected (green), mildly affected (orange), or strongly affected (red) by the dimerization mutation in different colours over a secondary structure prediction of the Nicn1 3’ UTR (Figure 5F). As can be seen from this analysis, no obvious structural or sequence constrain that would explain the observed differences can be identified. For instance, most sites, irrespective of their response towards the dimerization mutant are located opposite U residues and are in seemingly stable, double-stranded structures. Thus, in general we find that the dimerization mutant can reduce editing at specific sites, while not affecting editing levels at other sites.*”

4) *I am curious to know if this dsRBD3-mediated dimerization is conserved between different primates or mammals. Are the key residues conserved across various primates or mammals?*

We thank the reviewer for the question. Key residues for dimerization are absolutely conserved across mammals, and almost entirely conserved (with only 1 or 2 minor mutations) in all vertebrates down to *Polyodon spathula*. Non vertebrate organisms belonging to the Cephalochordata and Tunicata subphylums, which have ADARs with only one or two dsRBDs, do not retain the identified elements involved in dimerization (more than 6 mutations at key positions). We have now included an alignment with this information as new Supplementary Figure S12. In addition, we have added the following sentence in the discussion: “*Regarding the amino acid conservation of the key residues identified, dimerization of ADAR1-dsRBD3 is certainly conserved in mammals and most likely also conserved in all vertebrates from fish to mammals (Supplementary Figure S12).*”

Minor comments:

5) *How did the authors decide on the boundaries of dsRBD3-Long (residues 688-817) and dsRBD3-mid (residues 708-801)? Why 688-817 and 708-801?*

We agree with the reviewer that this point was not well explained in the manuscript. We selected the dsRBD3-mid construct (residues 708-801), for being a construct with the minimal folded part of dsRBD3 (716-797) and with middle size extensions on both sides. But the exact boundaries were chosen as such because it was a construct we previously studied, and that we found to be the minimal ADAR1 construct that retains nuclear localization properties (Barraud et al. PNAS 2014). For the dsRBD3-long construct, it was also a construct we previously studied for its localization properties, and that was here chosen such as to contain an additional ~15-20 residues on both sides of the dsRBD-mid construct. We have better explained this in the revised version of the manuscript.

6) *Acronyms should be spelled out in full on first use.*

Acronyms used in the manuscript were carefully checked and spelled out in full on first use in the revised version.

Reviewer #2

ADAR1 is an important enzyme in vertebrates that deaminates adenine bases to inosine in the context of dsRNA, and plays important roles in post-transcriptional RNA modification and innate cellular immunity. Mboukou et al present a timely structural and functional study of dimerization in ADAR1 by examining the homodimerization of the third dsRNA binding domain of this protein (dsRBD3), without and with dsRNA. This brings new data that weren't accessible in prior NMR studies of this domain and sheds a new perspective on the architecture of RNA editing proteins. This has important implications for understanding their mechanism in a cellular context.

By combining solution data (SAXS), crystal structures, co-immunoprecipitation and editing assays, the authors provide an excellent body of data from well-executed experiments to support their new insight that ADAR1 dsRBD3 forms a symmetrical homodimer. Moreover, they show, with a crystal structure, that this homodimer can bind two molecules of dsRNA in a back-to-back orientation. By differential tag co-immunoprecipitations, the authors show that mutations or deletions that impact dimerization in vitro can affect co-precipitation from cells. Finally, they demonstrate that loss of dimerization of dsRBD3 affects a subset of editing sites.

This is an important piece of work and should be published, albeit with some modifications to improve data presentation/interpretation and clarity for the reader.

We thank the reviewer for the positive feedback.

1. *Introduction, paragraph 2. The phenotypic data that are discussed relate to mice - this should be made clear in the text.*

We have made this point clearer in the text. The paragraph now starts with “*In mice, loss of ADAR1...*”

2. *Results paragraph 2: the buried surface area of the interface is denoted as 490 Å². The authors should clarify if this is the total buried surface area or the area per subunit.*

Evaluation of the buried surface area were initially conducted using PYMOL scripts. Thanks to the reviewer's question, we re-evaluated this aspect. In the revised manuscript, we now give the values reported by PISA (namely PDBePISA server <https://www.ebi.ac.uk/pdbe/pisa/>) for the PDB entry 7ZJ1. The values are slightly different from our initial evaluation. Values reported by PISA are: 575.7 Å² for the interface and 1151.4 Å² for the total buried area. This is now clearly stated in the manuscript: “*This interface has a surface area of ~575 Å², which corresponds to a total buried area of ~1150 Å²*”.

3. *The CRYSQL calculations show very convincingly that the dimer structure better represents the data than a monomer. It would be great if the authors could complement this by showing the fit of the crystal structure to a bead model of the dimer calculated e.g. by GASBOR.*

We thank the reviewer for this suggestion. We have now performed *ab initio* shape reconstruction using both the DAMMIN and GASBOR programs. Ten independent model building runs were performed in each case, and the *ab initio* models were aligned and analyzed using DAMAVER. Averaged bead models were also computed for the most populated cluster of each ensemble and

aligned with the crystal structure of dimeric ADAR1-dsRBD3. This new analysis is now presented in Supplementary Figure S3. The detailed procedure is described in the material and methods section. We also expanded the paragraph ‘The dimer observed in the crystal corresponds to the one existing in solution’ by adding the following text: *“In addition, we performed ab initio shape reconstructions from the dsRBD-short SEC-SAXS data using independently the DAMMIN and GASBOR programs. These ab initio dummy residue models further confirmed that the solution scattering data obtained in solution with ADAR1-dsRBD3 are perfectly compatible with the dimeric form observed in the crystal structure (Supplementary Figure S3).”*

4. *The SAXS experiments show some evidence of aggregation and the SEC-SAXS profiles are also not symmetrical, indicating non-ideal behavior. The range of images used for SAXS curve calculation from the SEC-SAXS experiments should be indicated on the chromatograms and the SAXS scattering profiles derived from these images should also be shown in the supplement.*

We agree that there is some aggregation and a non-ideal behavior of the ADAR1-dsRBD3 samples (see also response to point 6 below). We would just like to underline that we have not used these data to perform extensive modelling, but mostly to estimate the molecular weight of the various objects in solution. As suggested by the reviewer, we have now added the range of images used for SAXS curve calculation both in Table 1 (they were already present in Table 2), and on the chromatograms of Supplementary Figures S1 and S7 as gray background. We now also show the scattering profiles on Figures S1 and S7.

We also would like to mention that, while preparing the new figures with the range of images used to derive the SAXS curve, we realized that the automatic frame selection via synchrotron scripts was not ideal in the case of the dsRBD3-mid sample. For this sample, we thus ‘manually’ selected the ranges of images used, and re-evaluated the R_g , D_{max} and apparent molecular weight. The new values, with only minor changes compared with the initial values, are reported in the revised Table 1.

5. *It would be easier to compare the Kratky plots in the supplementary data if the authors plotted as normalised Kratky.*

We now show Normalized Kratky plots instead of Kratky plots for all SAXS data of Supplementary Figures S1 and S7.

6. *It’s rather surprising that the SEC-MALLS data show increasing Mw at later elution volumes. These SEC profiles are also non-symmetric. This suggests that there might be some non-ideal behavior, with the aggregated protein perhaps interacting with the matrix of the column. The limitations of these data should be discussed more fully.*

We agree with the reviewer and have now added the following text in the paragraph ‘Mutations at the interface render ADAR1-dsRBD3 monomeric’: *“We would like to point out that the studied samples display some non-ideal behaviour both in SEC-MALLS and SEC-SAXS, with a certain level of aggregation and probably some interaction with the SEC column, leading to non-symmetrical profiles. While such properties might prevent the use of SEC-SAXS data for accurate molecular modelling, the simple molecular weight estimates derived from these data can confidently be used to estimate whether the constructs are behaving as monomers or dimers in solution.”*

7. *Where the authors have used ITC data to show RNA binding in the dimeric dsRBD3 versus monomerising mutants, the difference in binding is quoted as “(factor 3.3 +/- 0.8)”. It’s not really clear how +/-0.8 is calculated for a factor or what it really means. I think it would be clearer to simply quote the K_d value of the WT ADAR3 dsRBD3 RNA binding here.*

We agree with the reviewer. We have replaced the factor with the actual value for the wild-type ADAR1-dsRBD3, i.e. $K_D = 125 \pm 25$ nM.

8. *The co-IP data are really important to the manuscript but this is the least well-executed component. It is not clear how many different experiments have been undertaken; this should be stated clearly. They should also clarify in the main body of the text that these experiments are done with full length proteins and use more informative labelling on in the figure i.e. ADAR1, rather than "A1".*

We agree with this reviewer that this part of the manuscript was not well described. We have now added the requested labeling to the figure, and also performed all experiments in triplicate. In the text, we now describe the use of full-length and truncated versions of ADAR1. A quantification of the wild-type and mutant co-IPs is now also added in Figure 4 (B and D and Supplementary Figure S10). Also, the text is now giving a better description of the performed experiments. We now state: *"This was performed in the presence of RNases to disrupt any RNA-mediated co-precipitation. All experiments were performed in triplicate and quantified (Figure 4B, D)."*

9. *The co-IPs also show some variation in the transfection efficiency (annoying, but a common issue) as well as the efficiency of the pull-down (not ideal). This makes it a bit difficult to compare across samples in these experiments. Can the authors provide some quantification of the co-IPed HA-tagged protein bands versus the IPed FLAG-tagged protein bands? This would make the differences between the ADAR1 dsRBD3 mutations versus the delta-deaminase constructs clearer.*

We do agree with this comment. The quantified bar graphs with standard deviations are now incorporated into Figure 4 (B and D). The additional blots used for quantification can be found in the Supplementary Figure S10.

10. *The editing assays are a further important validation of the functional importance of the dsRBD3 dimerization. There seems to be quite a bit of variation depending on whether sites are mostly edited by one isoform or another of ADAR1. The discussion could be briefly expanded to highlight that data for some sites is noisier than others.*

We agree with this comment. We have tested a set of additional targets and have now included a graph depicting the relative changes observed in the 3' UTR of Nicn1 which is heavily edited. However, as can be appreciated from Figure 5F, there seems no obvious logic by which a particular site is affected or not affected by the dimerization mutation. We have now also discussed this point in more detail. See also the detailed response to point 4 raised by Reviewer #1.

11. *The new structure of ADAR1 dsRBD3 alters our understanding of prior models of dsRBD3 contributing to nuclear import via interactions with transportin. Given that previous models were based on limited data and that there are now better modelling tools available, could the authors re-evaluate/regenerate the model with transportin they presented in Barraud et al 2014?*

REDACTED

As suggested by the present reviewer, we also had the idea of using advanced modelling tools, and we tried to use AlphaFold-Multimer (both as a standalone program or via ColabFold) to model structure of the complex between Trn1 and ADAR1-dsRBD3 monomer or dimer (one or two copies as input). (We also have to mention that AlphaFold-Multimer failed at predicting the ADAR1 dsRBD3/dsRBD3 dimer when tried in isolation - with the predicted template modelling score at the interface (ipTM score) < 0.35 for all models - and not displaying anything close to the crystal structure we solved). In the two cases (dsRBD3 as a monomer or a dimer), AlphaFold-Multimer was not confident at all in predicting the interaction interface with Trn1 (with ipTM scores < 0.31 for the 'monomer' models and < 0.27 for the 'dimer' models).

In addition, we had also tried to re-evaluate our previous model presented in Barraud et al. 2014. And, even though, generally speaking, it could seem feasible for ADAR1-dsRBD3 to bind to Trn1 as a dimer, since the interaction interface (N- and C-terminal flanking modules on the alpha-helical side) is opposite to the dimerization interface (the beta-sheet surface), the structural model constructed in Barraud et al. 2014 is not directly compatible with the binding of a dimer of ADAR1-dsRBD3 to Trn1 (several clashes of the second ADAR1-dsRBD3 monomer exist with Trn1's HEAT repeats 6 and 7). With manual adjustments, it seems possible that small domain reorientation, without changing drastically the contacts at the interface between ADAR1-dsRBD3 and Trn1, could allow a second dsRBD to fit at the exit of the C-terminal arch. However, since we do not know whether ADAR1-dsRBD3 binds to Trn1 as a monomer or as a dimer, we have not explored this possibility further.

In the text, we now present these aspects in the discussion part:

“Although only dimeric forms of ADAR1-dsRBD3 were observed in vitro in our study, it would be interesting to know whether the ADAR1-dsRBD3 dimer can dissociate to form monomers under certain conditions. For example, it has been reported that the Dicer/ADAR1 complex is likely to consist of one Dicer and one ADAR1 molecule, but not an ADAR1 homodimer (Ota et al. 2013). The ADAR1 homodimer could therefore dissociate upon binding to other protein partners. This possibility is particularly relevant in the case of ADAR1-dsRBD3 interaction with the nuclear import receptor Trn1. In fact, we previously constructed a model of the interaction between Trn1 and ADAR1-dsRBD3 as a monomer, since its dimerization capabilities were unknown at the time (Barraud et al. 2014). Although it might seem structurally feasible for ADAR1-dsRBD3 to bind to Trn1 as a dimer, since the interaction interface is basically opposite to the dimerization interface, the structural model we previously constructed is not directly compatible with the binding of a dimer of ADAR1-dsRBD3 to Trn1. There are indeed several clashes between the second ADAR1-dsRBD3 monomer and Trn1 in this overall organisation. From a structural point of view, ADAR1-dsRBD3 should either bind to Trn1 with a different orientation, allowing a second dsRBD to fit at the exit of the Trn1 concave pocket, or bind as a monomer, with dissociation of the dimer that could be favoured upon binding to Trn1. Further studies are required to distinguish between these possibilities.”